# Mathematical Modeling of ROS Production and Diode-like Behavior in the SDHA/SDHB Subcomplex of Succinate Dehydrogenases in Reverse Quinol-Fumarate Reductase Direction

**DOI:** 10.3390/ijms232415596

**Published:** 2022-12-09

**Authors:** Nikolay I. Markevich, Lubov N. Markevich

**Affiliations:** 1Institute of Theoretical and Experimental Biophysics of RAS, 142290 Pushchino, Russia; 2Institute of Cell Biophysics of RAS, 142290 Pushchino, Russia

**Keywords:** succinate dehydrogenase (SDH), a tunnel-diode behaviour, fumarate reduction, complex II, reactive oxygen species (ROS), computational model

## Abstract

Succinate dehydrogenase (SDH) plays an important role in reverse electron transfer during hypoxia/anoxia, in particular, in ischemia, when blood supply to an organ is disrupted, and oxygen is not available. It was detected in the voltammetry studies about three decades ago that the SDHA/SDHB subcomplex of SDH can have such a strong nonlinear property as a “tunnel-diode” behavior in reverse quinol-fumarate reductase direction. The molecular and kinetic mechanisms of this phenomenon, that is, a strong drop in the rate of fumarate reduction as the driving force is increased, are still unclear. In order to account for this property of SDH, we developed and analyzed a mechanistic computational model of reverse electron transfer in the SDHA/SDHB subcomplex of SDH. It was shown that a decrease in the rate of succinate release from the active center during fumarate reduction quantitatively explains the experimentally observed tunnel-diode behavior in SDH and threshold values of the electrode potential of about −80 mV. Computational analysis of ROS production in the SDHA/SDHB subcomplex of SDH during reverse electron transfer predicts that the rate of ROS production decreases when the tunnel-diode behavior appears. These results predict a low rate of ROS production by the SDHA/SDHB subcomplex of SDH during ischemia.

## 1. Introduction

Succinate dehydrogenase (SDH) is one of the key enzymes of cell energy metabolism, linking the Krebs cycle and the electron transport chain (ETC). SDH reversibly oxidizes succinate to fumarate and transfers the electrons produced by this reaction to the membrane quinone pool, providing ubiquinol QH_2_ for oxidative phosphorylation in the cell. SDH can have strong nonlinear properties in both directions, such as hysteresis and bistability in forward succinate-quinone oxidoreductase activity [1] and a “tunnel-diode” behavior in reverse quinol-fumarate reductase direction [2]. Although the “tunnel-diode” behavior, that is, a strong drop in the rate of fumarate reduction as the driving force (over potential) is increased, has been detected in the voltammetry studies about three decades ago [2], however, molecular and kinetic mechanisms of this phenomenon remain unclear and studied in this work.

It is important to note that the “tunnel-diode” behavior in the reverse electron transfer in complex II is observed at the level of the hydrophilic SDHA/SDHB subcomplex. Figure 1 schematically shows the reverse electron transfer in complex II. In this case, the electron donor is quinol QH_2_ in the CoQ (quinone-binding) site localized in the hydrophobic SDHC/SDHD subcomplex in the inner mitochondrial membrane. Two electrons are sequentially transferred, one after the other, to the soluble SDHA/SDHB subcomplex exposed on the matrix side of the mitochondrial inner membrane. First, the electrons are transferred to the chain of [Fe-S] clusters located in the SDHB subunit and then to the FAD-containing SDHA subunit, where the fumarate is reduced to succinate.

Reverse electron transfer in SDH occurs in conditions of oxygen deficiency (hypoxia/anoxia). It was found [3] that complex I (CI) and dihydroorotate dehydrogenase (DHODH) can still deposit electrons into ETC when oxygen is not available as the terminal electron acceptor of the respiratory chain. In this case, cells lacking oxygen reduction accumulate ubiquinol, driving SDH in the reverse direction and depositing electrons onto fumarate, resulting in succinate generation [3] which may be very important during ischemia.

It was proposed that in ischemia when blood supply to an organ is disrupted and oxygen is not available, reversal of mitochondrial complex II (CII) results in ischemic succinate accumulation, that is, ubiquinone Q reduction by CI driving reversal of CII, with fumarate as an electron acceptor yielding succinate oxidation that drives injury at reperfusion [4]. However, the mechanism of experimentally observed ischemic succinate accumulation is controversial [5]. It was shown recently [6] that CII reversal is possible in hypoxic mitochondria but is not the primary succinate source in hypoxic cardiomyocytes or ischemic hearts. That is, ischemic succinate is generated by canonical Krebs cycle activity mostly from α-ketoglutarate and upstream metabolites rather than by mitochondrial CII reversal [6]. A detailed analysis of different models of ischemic succinate accumulation was performed by Chinopoulos in the review [5]. In particular, it was noted that in ischemia, SDH reverses, forming succinate only minorly due to a diode-like property. And besides, mammalian mitochondria lack suitable quinones that could support SDH reversal.

Thus, the main goal of this work was not only to analyze changes in the kinetics of electron transfer responsible for the diode-like behavior of SDH in reverse quinol-fumarate reductase direction but also to study conditions when diode-like behavior is observed. 

The authors who first discovered the tunnel-diode behavior [2] suggested that this phenomenon may be based on a strong decrease in the binding of fumarate or release of succinate when electron transfer is accelerated in the opposite direction, that is, fumarate reduction. We tested this hypothesis by analyzing computer-simulated steady-state rates of reverse electron transfer in the SDHA/SDHB subunits of SDH with a change in the binding/release constants of fumarate/succinate with the active center of SDH during the fumarate reductase reaction and analyzed a dependence of threshold values of the electrode potential when diode-like behavior is observed on different kinetic parameters and concentration of reduced/oxidized redox centers of CII.

Besides, we analyzed ROS production in the SDHA/SDHB subcomplex of SDH during the reverse quinol-fumarate reductase activity and predicted that the rate of ROS production decreases at a decrease in the rate of succinate release from the active site of SDH when FADH^•^ and FADH_2_ are proposed as the major ROS producing redox centers. These results predict a low rate of ROS production by the SDHA/SDHB subcomplex of SDH during hypoxia/anoxia, that is, during ischemia.

## 2. Results and Discussion

### 2.1. E_out_ Dependency of the Total Rate of Reverse Electron Transfer in SDH

Steady-state E_out_ dependencies of the total rate of reverse electron transfer in the SDHA/SDHB subcomplex of SDH at different values of the rate constants of fumarate and succinate binding to the dicarboxylate binding site upon FAD reduction are presented in Figure 2. The total rate of reverse electron transfer (V_rev_tot_) in SDH is computed as the sum of rates of oxidation of the cluster [2Fe-2S] in reactions 5, 6, 9, 11, 13, 15 and equal to V_rev_tot_ = V_5_ + V_6_ + V_9_ + V_11_ + V_13_ + V_15_. It should point out that V_rev_tot_ = 2∙V_1_ = 2∙V_2_ = 2∙V_3_ in the steady state because two electrons are transferred to one succinate molecule. Here, V_1_, V_2_, and V_3_ are the rates of reactions 1, 2, and 3, respectively.

Figure 2A shows that when a decrease in constants K_eq10_ and K_eq12_ occurs due to a decrease in k_on_ rate constants, k_10_ and k_12_, a tunnel-diode behavior is observed, but only at very high negative threshold values of the electrode potential E_out_ (about −200 mV), which is significantly higher than the experimentally observed values (about −80 mV [2]). If a decrease in constants K_eq10_ and K_eq12_ occurs due to only an increase in k_off_ rate constants, k_-10_ and k_-12_ (Figure 2B), then tunnel-diode behavior is completely absent.

On the contrary, Figure 2C,D shows a decrease in the rate of dissociation of succinate from the active center at reduced FAD forms, that is, a decrease in equilibrium constants K_eq14_ and K_eq16_ due to a decrease in the rate constants k_14_ and k_16_ (Figure 2C) or an increase in the rate constants k_−14_ and k_−16_ (Figure 2D) upon FAD reduction results in a strong tunnel-diode behavior. Moreover, the threshold potential may be close to the experimentally observed values of about −60 [7] or −80 [2] mV at k_14_ = k_16_ = 10^−2^ s^−1^, K_eq13_ = 0.24 µM^−1^, K_eq14_ = K_eq16_ = 0.25 µM (curve 3 in Figure 2C).

It is important to note here that a decrease in K_eq14_ and K_eq16_ is accompanied by a simultaneous increase in K_eq13_ according to the “detailed balance” relations [8], i.e., K_eq13_ = K_eq8_ ∙ K_eq9_/K_eq14_. This is surprisingly consistent with the experimentally observed [9] strong increase in the midpoint redox potential of the FAD/FADH· pair when succinate binds to SDH. The midpoint potential of the cluster [2Fe-2S] also increases with succinate binding, but to a much lesser extent [9], so that leads to an increase in the constant equilibrium K_eq13_.

### 2.2. E_out_ Dependency of the Rate of Succinate Production in SDH

In order to better understand the mechanism of the tunnel behavior in SDH in the reverse direction, it is convenient to analyze the steady-state E_out_ dependence of the rates V_8_, V_14_, and V_16_ at different values of the binding/release constants of fumarate/succinate with the active center of SDH. These dependencies are presented in Figure 3, Figure 4, Figure 5 and Figure 6.

It should be noted that the total current in the SDH in the reverse direction is exactly equal to twice the total rate of free succinate production because two electrons are transferred during the formation of one succinate molecule, which is V_rev_tot_ = 2∙V_suc_tot_, where V_suc_tot_ = V_8_ + V_14_ + V_16_. 

The steady-state E_out_ dependencies of the rates V_8_, V_14_, and V_16_ at different values of the succinate release constants from the active center of SDH are presented in Figure 3. First of all, it should be emphasized that the electron transfer rate constants from the cluster [2Fe-2S] to FAD are several orders of magnitude higher than the dissociation constants of succinate. Therefore, electron transfer rates, V_13_ and V_15_, significantly prevail over the V_8_ and V_14_ rates, respectively, at the branching points with succinate dissociation, such as FAD.suc and FADH·.suc. This means that an increase in the rates of reactions 13 and 15 inhibits the succinate dissociation in reactions 8 and 14, respectively.

Figure 3A shows that the E_out_ dependence of the succinate dissociation rate V_8_ (curve 2) has a nonmonotonic behavior at the basal values of all equilibrium constants. That is, it decreases when the electrode potential E_out_ decreases below a certain threshold value. This drop is due to the strong predominance of the electron transfer rate from the cluster [2Fe-2S] to FAD, V_13_, over the rate V_8_ at very low E_out_ values. However, the total succinate production rate V_suc_tot_ (curve 1) increases monotonically with a decrease in E_out_ due to a strong increase in the dissociation rate V_16_ (curve 3). The rate V_14_ (curve 4) always remains low due to the high rate V_15_.

The 10-fold decrease in succinate dissociation constants k_14_ and k_16_ from 10 to 1 s^−1^ (Figure 3B) results in a strong nonmonotonic (a tunnel-diode behavior) in the E_out_ dependence of the rate V_suc_tot_ (curve 1). This happens because the succinate dissociation rate V_16_ becomes very low (curve 3) and cannot compensate for the drop in the rate V_8_ (curve 2) when E_out_ decreases below the threshold value. The rate V_14_ (curve 4) remains very low. With a further decrease in the constants k_14_ and k_16_ from 10 to 0.1 (Figure 3C) and 0.01 s^−1^ (Figure 3D), the qualitative behavior of the rates V_suc_tot_, V_8_, V_14_, and V_16_ is the same as in the previous case, only a drop in V_suc_tot_ begins at more positive E_out_ values up to −60 mV observed in experiments [7]. Moreover, the total succinate production rate V_suc_tot_ is completely determined only by the rate V_8_ (curves 1 and 2).

Thus, the results of computer simulation of a decrease in the rate constants of succinate dissociation k_14_ and k_16_ and, correspondingly, in K_eq14_ and K_eq16_, taking into account simultaneous increase in K_eq13_, can well quantitatively explain the experimentally observed tunnel-diode behaviour in the reverse transfer of electrons to SDH.

Quite different properties of the appearance of nonmonotonicity in the rate V_suc_tot_ are observed when the equilibrium constants K_eq14_ and K_eq16_ decrease due to an increase in the constants k_−14_ and k_−16_ of the succinate binding to the active center of SDH (Figure 4). Figure 4A is identical to Figure 3A. First of all, k_−14_ and k_−16_ must be increased by 1000 times compared to the basal value of 0.04 µM^−1^ s^−1^, i.e., up to 40 µM^−1^ s^−1^ in order for nonmonotonicity to appear in the E_out_ dependence of the V_suc_tot_ (Figure 4B). Just as in the previous case, the drop in the rate V_suc_tot_ (curve 1) when E_out_ decreases below the threshold value occur due to a decrease in the rate V_16_ (curve 3), which is no longer able to compensate for the drop in the rate V_8_ (curve 2). There is a further decrease in the rates V_8_ (curve 2) and V_16_ (curve 3) to almost 0 with E_out_ less than the threshold value when k_−14_ and k_−16_ increase up to 400 (Figure 4C) and 4000 µM^−1^ s^−1^ (Figure 4D). In this case, the total rate of succinate release, V_suc_tot_, is determined only by the rate V_14_ (curves 1 and 4).

As can be seen from Figure 4D (see also curve 4 in Figure 2D), the E_out_ threshold values can approach the experimentally observed values of about −60 mV. However, the values of the equilibrium constant K_eq13_ satisfying the “detailed balance” relations have to be very high K_eq13_ = 24 µM^−1^ at very high rate constants k_−14_ and k_−16_ and do not correspond to the experimentally observed values of midpoint potentials E_m_(FAD.suc/FADH^•^.suc) and E_m_([2Fe-2S]).

As for the change in the succinate generation rates with a decrease in the binding of fumarate to the active center of SDH during FAD reduction, which corresponds to the data on reverse electron transfer in Figure 2A,B, these data are presented in Figure 5 and Figure 6. Computer simulation analysis shows that if a decrease in the equilibrium constants K_eq10_ and K_eq12_ occurs only due to a decrease in the fumarate binding constants k_10_ and k_12_, then the nonmonotonicity in the E_out_ dependence of V_suc_tot_ is explained as well as in previous cases by a strong drop in the rate V_16_ (Figure 5). However, the threshold potential, in this case, is about −200 mV and is very far from the experimentally observed values of about −60 [7] or −80 [2] mV.

If the equilibrium constants K_eq10_ and K_eq12_ decrease only by increasing the reverse fumarate release constants k_-10_ and k_-12_, then nonmonotonicity in total succinate production, i.e., the tunnel effect, is completely absent (Figure 6).

### 2.3. Computational Analysis of the ROS Production Rate in the SDHA/SDHB Subcomplex during Reverse Electron Transfer

Initially, it was analyzed dependence of the concentration of different redox centers of the SDHA/SDHB subcomplex in the reduced state on E_out_. This is important for at least two reasons. First, we have to understand how the concentration of ROS-producing redox centers in the reduced state changes with a change in E_out_. And secondly, it is necessary to know the interdependence of E_out_ and the concentration of the cluster [3Fe-4S] in the reduced state in order to extrapolate the E_out_ dependence of various rates shown in Figure 2, Figure 3, Figure 4, Figure 5 and Figure 6 on the dependence of the same rates on the concentration of the reduced cluster [3Fe-4S]. Then we will be able to predict the possibility of achieving a tunnel effect in real physiological conditions.

The E_out_ dependence of the concentration of FADH^•^ and FADH_2_, as well as the cluster [3Fe-4S] in the reduced state is shown in Figure 7. These redox centers were previously proposed [10,11,12] as the main ROS generators in the subcomplex SDHA/SDHB of SDH. As can be seen from the results presented in Figure 7, only changes in the concentration of FADH^•^ on E_out_ have nonmonotonic dependence (Figure 7A,B). Moreover, it is interesting that these nonmonotonic changes are only at a simultaneous decrease in the rate constants, k_14_ and k_16_, of succinate dissociation from reduced FAD forms (Figure 7B) can have the experimentally observed values of the threshold potential of about −60 [7] or −80 [2] mV. It is important to note that the concentration of FADH^•^ and FADH_2_ decreases with a decrease in the rate of release of succinate from the active center, that is, with a decrease in the equilibrium constants K_eq14_ and K_eq16_ (Figure 7B,D). It should also be noted an important property of the dependence of the concentration of the reduced cluster [3Fe-4S], [3Fe-4S]^−^, on E_out_ (Figure 7E). The concentration of [3Fe-4S]^−^ reaches values close to the maximal value of 235 µM already at E_out_ about −60 mV. This means that the “tunnel-diode” behavior in reverse quinol-fumarate reductase direction observed experimentally at −60 mV is possible only with maximal [3Fe-4S] cluster reduction, which is apparently difficult to achieve under physiological conditions. At the same time, the dependence of the concentration of the reduced cluster [2Fe-2S]^−^ on E_out_ (Figure 7F) is much weaker and reaches a maximum value of 235 µM at values of E_out_ about −200 mV. Net reversal of the mammalian SDH complex has been considered thermodynamically unfavorable because the standard reduction potential of ubiquinone (UQ) is slightly greater than that of fumarate [7]. However, because the reduction potential of UQ and fumarate are very close to each other (~10 mV apart), it was shown that UQH_2_ accumulation drives the net reversal of the SDH complex in mammalian cells upon suppression of O_2_ reduction [3]. 

The E_out_ dependence of the stationary rates of ROS production is presented in Figure 8. The total stationary rate of H_2_O_2_ production, VH_2_O_2_, was computed as the rate of H_2_O_2_ release from the mitochondrial matrix to the cytosol, V_22_, that equal to the summary rate of H_2_O_2_ production by FADH_2_, V_17_, and dismutation of O_2_∙^−^, V_21_, in the matrix at the steady state (see Appendix A). Values of the catalytic constants of ROS formation by different redox centers in the subcomplex SDHA/SDHB were taken from our previous computational model of CII in the forward succinate-quinone oxidoreductase (SQR) direction [13]. That model was calibrated by fitting the computer-simulated results to experimental data obtained on submitochondrial particles prepared from bovine [14] and rat heart [11] mitochondria upon inhibition of the Q-binding site by atpenin A5 and Complex III (CIII) by myxothiazol, respectively. It was shown [13] that only reduced flavin adenine dinucleotide (FADH_2_) in the unoccupied dicarboxylate state and flavin semiquinone radical (FADH^•^) feature the experimentally observed bell-shaped dependence of the rate of ROS production on the succinate concentration upon inhibition of CIII or Q-binding site of CII, i.e., suppression of SQR activity. At the same time, the maximal rate of ROS production was about 1000 pmol/min/mg mitochondrial protein. 

As we noted in Appendix A of this work, in order to compare computer-simulated rates of ROS production presented in Figure 8 in μM/s with experimentally observed rates expressed in pmol/min/mg protein the computer-simulated rates should be multiplied by a factor of 220, i.e., 1 μM/s = 220 pmol/min/mg mitochondrial protein. As the simulation results show, only a simultaneous decrease in equilibrium constants K_eq10_ and K_eq12_ of binding of fumarate to the dicarboxylate binding site upon FAD reduction due to a decrease in k_on_ rate constants, k_10_ and k_12_, (Figure 8A) can result in very high rates of ROS production more than 1000 pmol/min/mg mitochondrial protein. That is clear because the decrease in the rate constants k_10_ and k_12_ results in an increase in the concentration of the major ROS-producing redox centers FADH^•^ and FADH_2_. However, our preliminary analysis shows that a decrease in the rate constants of fumarate binding to the dicarboxylate binding site upon FAD reduction, k_10_ and k_12_, is unlikely to account for experimentally observed values of the threshold potential during the “tunnel-diode” behavior and should not be considered as a possible high rate of ROS production at reverse SDH activity. All other changes in the rate constants of binding or release of succinate/fumarate have a small effect on the rate of ROS production by the subcomplex SDHA/SDHB in the reverse direction of SDH (Figure 8B–D). Thus, a decrease in the rate of succinate release from the active center during the reduction in FAD to FADH_2_ that quantitatively explains the experimentally observed tunnel-diode behavior in SDH from the beef heart [2] and *Escherichia coli* [7] mitochondria results in a very low rate of ROS production in the reverse direction of SDH (Figure 8C,D) when FADH^•^ and FADH_2_ are proposed as the major ROS producing redox centers.

However, it should be pointed out the recent work [12] in which it was shown that in the absence of respiratory chain inhibitors, the model analysis revealed the [3Fe-4S] iron-sulfur cluster as the primary O_2_∙^−^ source. In this case, taking into account the very high concentration of [3Fe-4S]^−^ at relatively small values of E_out_ as shown in Figure 8E, it is very likely to expect a high rate of ROS production by this cluster [3Fe-4S]^−^ of SDH during a “tunnel-diode” behavior in reverse quinol-fumarate reductase direction in the absence of respiratory chain inhibitors. Thus, real changes in the rate of ROS production by the subcomplex SDHA/SDHB in the reverse direction of SDH during hypoxia/anoxia, that is, during ischemia, depends on the real experimental condition.

### 2.4. Dependence of the Rate of Succinate Production on the Fumarate Concentration in SDH

Computer-simulated dependencies of the stationary rates of succinate production in the reverse direction of SDH on the fumarate concentration at different values of the electrode potential E_out_ are presented in Figure 9A–D. Figures A–D differ in the values of the equilibrium constants K_eq13_, K_eq14_, and K_eq16_, which change due to a decrease in the rate constants of succinate release from the active center k_14_ and k_16_ from high control values (Figure 9A) to very small values (Figure 9D) when electron transfer is accelerated in the opposite direction. As can be seen from Figure 9, the dependence of the total succinate generation rate on the fumarate concentration has the usual hyperbolic character for any values of E_out_ and equilibrium constants K_eq13_, K_eq14_, and K_eq16_. At the same time, with a decrease in the values of the rate constants k_14_ and k_16_, the maximum rate of succinate production begins to decrease when the negative values of E_out_ exceed some threshold values (about −200 mV in Figure 9B), which the less in absolute value (about −150 and −100 mV), the less the rate constants of succinate release from the active center in Figure 9C,D, respectively.

## 3. Methods and Models

### 3.1. Kinetic Model of Reverse Electron Transfer in SDHA/SDHB Subunits of SDH

A kinetic scheme of reverse electron transfer and O_2_∙^−^/H_2_O_2_ production underlying a mechanistic computational model of SDH activity in the reverse direction (fumarate reduction) in subunits SDHA and SDHB is presented in Figure 10. This kinetic scheme includes the following redox centers: (a) three iron-sulfur clusters: [2Fe-2S], [4Fe-4S], and [3Fe-4S]) located in SDHB subunit (Figure 10A) that transfer electrons one at a time to produce succinate in the SDHA subunit; (b) flavin adenine dinucleotide, FAD, located in SDHA subunit (Figure 10B).

Reverse electron transfer reactions in the SDHA/SDHB subunits of SDH include both the mainstream electron pathway from [3Fe-4S] cluster to fumarate and bypass reactions resulting in O_2_∙^−^/H_2_O_2_ formation. These bypass reactions are marked in red in the kinetic scheme (Figure 10C).

This scheme is supported by the numerous literature data on electron transfer pathways between different redox centers of SDH [10,11,15,16]. 

The entire reaction network of electron transfer and O_2_∙^−^/H_2_O_2_ production corresponding to this kinetic scheme in Figure 10 consists of 22 reactions that are described in detail in Table 1. Midpoint redox potentials, rate constants, and concentrations are taken from the experimental data (see Table 2 and references therein). 

The reverse electron transfer in the soluble subcomplex SDHA/SDHB begins with the reduction in the cluster [3Fe-4S] (reaction 1) in the SDHB subunit with subsequent single electron transfer from the [3Fe-4S] cluster through [4Fe-4S] iron-sulfur center (reaction 2) to the cluster [2Fe-2S] (reaction 3) as shown in Figure 10A. The initial reduction in the cluster [3Fe-4S] (reaction 1) can be carried out by various electron donors, the nature of which depends on experimental conditions. In physiological conditions, electrons in reaction 1 are exchanged with the quinone pool. A detailed kinetic scheme of the exchange of the first and second electrons between the [3Fe-4S] cluster and the quinone pool in CII is presented in work [1]. In voltammetry experiments [2,7], electrons are exchanged directly between SDH and a graphite electrode. The rate of this exchange is controlled by the electrode potential (E_out_) and is considered in detail in Table 1 and Table 2. 

Figure 10B presents a kinetic scheme of chemical reactions of the reversible oxidoreduction of succinate, fumarate, and FAD, catalyzed by the SDH flavoprotein subunit A (SDHA). These reactions involve binding/dissociation of fumarate/succinate to/from the dicarboxylate binding site of the SDHA subunit as well as a single electron transfer from the iron-sulfur cluster [2Fe-2S] of the hydrophilic SDHB subunit to FAD in the SDHA subunit. Reactions of the reactive oxygen species formation in the soluble subcomplex SDHA/SDHB of SDH are presented in Figure 10C. 

The reduction in fumarate to succinate in the flavoprotein subunit SDHA shown in Figure 10B, can occur via various alternative pathways. Fumarate can first bind to the dicarboxylate binding site of SDH when FAD is oxidized (reaction 4), followed by the transfer of the first (reaction 5) and second (reaction 6) electrons from the cluster [2Fe-2S] to FAD with the formation of the complexes FADH^•^.fum and FADH2.fum, respectively. In reaction 7, fumarate is reduced to succinate with simultaneous oxidation of FADH_2_ to FAD. Then, in reaction 8, the FAD. suc complex dissociates with the release of succinate.

In another pathway of reducing fumarate to succinate, FAD in the unoccupied dicarboxylate state is sequentially reduced first to FADH^•^ in reaction 9 and then to FADH_2_ in reaction 11 receiving the first and second electrons, respectively, from the cluster [2Fe-2S]. In this case, fumarate can bind to the active site when FAD is in FADH^•^ (reaction 10) or FADH_2_ (reaction 12) state, respectively. After that, fumarate reduction to succinate and succinate release occur as well as in the previous pathway that is in reactions 6–8 or 7, 8.

In addition, reactions 13–16 represent the third pathway of two electrons transfer from the cluster [2Fe-2S] to the FAD and the release of succinate when succinate is initially bound to the active center of SDH.

These three pathways of formation of succinate from fumarate presented in Figure 2B include 4 thermodynamic cycles in which the initial and final states are identical (reactions 4–8; 6, 10–12; 8, 9, 13, 14; 7, 12, 13, 15, 16). Therefore, the equilibrium constants of the reactions along any cycle must satisfy so-called “detailed balance” relationships [8]. These detailed balance relations require the product of the equilibrium constants along a cycle to be equal to 1, as at equilibrium, the net flux through any cycle vanishes. Therefore, such relations decrease the number of independent rate constants in a kinetic model and imply that when any one of the equilibrium constants in any cycle changes, other constants in this cycle should automatically change. 

The restrictions on the kinetic constants of the reactions presented in Figure 10B are the following
(1)K_eq4_ ∙ K_eq5_ ∙ K_eq6_ ∙K_eq7_ ∙ K_eq8_ = 1;(2)K_eq6_ ∙ K_eq10_/(K_eq11_ ∙ K_eq12_) = 1;(3)K_eq8_ ∙ K_eq9_/(K_eq13_ ∙ K_eq14_) = 1;(4)K_eq7_ ∙ K_eq12_ ∙ K_eq13_ ∙K_eq15_ ∙ K_eq16_ = 1.

Potential redox centers of reactive oxygen species (ROS) generation in the soluble subcomplex SDHA/SDHB are FADH^•^, FADH_2_, and [3Fe-4S] are presented in Figure 10C. These redox centers were previously proposed [10,11,12] as the main ROS generators in the subcomplex SDHA/SDHB of SDH. 

FADH_2_ can generate either H_2_O_2_ in reaction 17 or superoxide O_2_·^−^ in reaction 18. The FADH^•^ radical can generate only superoxide O_2_·^−^ in reaction 19. The [3Fe-4S] cluster generates superoxide in reaction 20. In addition, intramitochondrial superoxide anion dismutation is represented by reaction 21.

### 3.2. Computational Model of Reverse Electron Transfer in SDHA/SDHB Subunits of SDH. Mathematical Model

A computational model consisting of 13 ordinary differential equations (ODEs) for the kinetic scheme presented in Figure 2, plus 4 moiety conservation equations, was derived from the reaction networks using the law of mass action and Michaelis (more exactly: Henri-Michaelis-Menten [25]) equation for all 22 kinetic processes. The models were implemented in DBSolve Optimum software available at the website http://insysbio.ru, accessed on 6 December 2022. The details of the mathematical model describing oxidized and reduced states of different carriers and electron flow through SDHA/SDHB subunits of SDH are presented in Appendix A. Values of model parameters, rate constants, and concentration of different electron carriers were taken from experimental data in the literature on thermodynamics and kinetics of electron transfer in CII of the respiratory chain (Table 1 and Table 2). Additionally, the model is presented in SBML format in a separate file: Appendix A.

## 4. Conclusions

A mechanistic computational model of reverse electron transfer in the subcomplex SDHA/SDHB of SDH was developed to account for a “tunnel-diode” behavior in reverse quinol-fumarate reductase direction in the SDHA/SDHB subcomplex detected in the voltammetry studies about three decades ago [2]. The model consists of a system of 13 ordinary differential equations that describe the dependence of the rates of electron flows and succinate production in SDH on the electrode potential E_out_. It was found that the “tunnel-diode” behavior in the SDHA/SDHB subcomplex of SDH, that is, a strong drop in the rate of fumarate reduction as the driving force is increased, is accounted for by a decrease in succinate release from the active site of SDH during reduction in flavin adenine dinucleotide, FAD, to FADH_2_. In particular, the model simulation predicts that experimentally observed threshold values of the electrode potential of about −60 [7] or −80 [2] mV can be explained quantitatively by a decrease in the rate constants of succinate release, k_14_, and k_16_, from the control value of 10 s^−1^ (curve 1 in Figure 2C) to k_14_ = k_16_ = 10^−2^ s^−1^ (curve 4 in Figure 2C).

It is important to emphasize that a decrease in dissociation constants of succinate from the active site is accompanied by a simultaneous shift of equilibrium from FAD to FADH· according to the “detailed balance” relations [8] in thermodynamic cycles of succinate formation and release as we noted in Section 3.1. “E_out_ dependency of the total rate of reverse electron transfer in SDH.” This is surprisingly consistent with the experimentally observed [9] strong increase in the midpoint redox potential of the FAD/FADH· pair when succinate binds to SDH.

Computational analysis of ROS production in the SDHA/SDHB subcomplex of SDH during the reverse quinol-fumarate reductase activity predicts that the rate of ROS production decreases at a decrease in the rate of succinate release from the active site of SDH when FADH^•^ and FADH_2_ are proposed as the major ROS producing redox centers. The concentration of FADH^•^ and FADH_2_ decreases with a decrease in the equilibrium constants K_eq14_ and K_eq16_ as shown in Figure 7B,D. However, the concentration of the reduced cluster [3Fe-4S]^−^ of SDH is maximal during a “tunnel-diode” behavior in reverse quinol-fumarate reductase direction, so it is very likely to expect a high rate of ROS production by this cluster if [3Fe-4S] is the primary O_2_∙^−^ source redox center in the absence of respiratory chain inhibitors as proposed in [12].

As to the participation of reverse electron transfer in SDH in ischemic succinate accumulation, it seems that it depends on real physiological conditions since SDH can be a major succinate source in ischemic in some cases [4]. However, sometimes ischemic succinate is generated by canonical Krebs cycle activity mostly from α-ketoglutarate and upstream metabolites rather than by mitochondrial CII reversal [6]. Our computational analysis predicts that a “tunnel-diode” behavior in the reverse quinol-fumarate reductase direction is observed only at the maximal reduction in the [3Fe-4S] cluster. So, it is unlikely that a tunnel effect can block ischemic succinate accumulation, as proposed in the review [5]. It seems more likely, that complex I and dihydroorotate dehydrogenase can support reverse electron transfer in SDH when oxygen is not available as the terminal electron acceptor of the respiratory chain, as shown in [3].

## Figures and Tables

**Figure 1 ijms-23-15596-f001:**
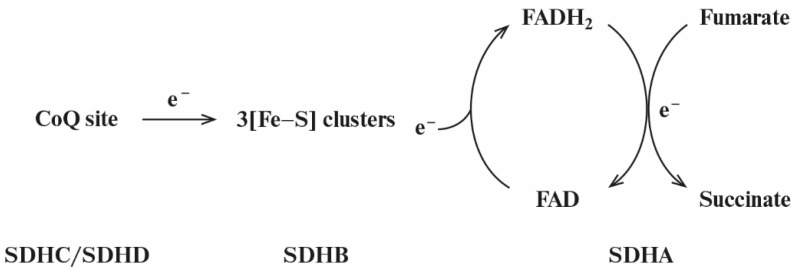
Simplified general scheme of reverse electron transfer in complex II from the CoQ site to the three [Fe-S] clusters located in SDHB subunit and further to the flavoprotein subunit SDHA.

**Figure 2 ijms-23-15596-f002:**
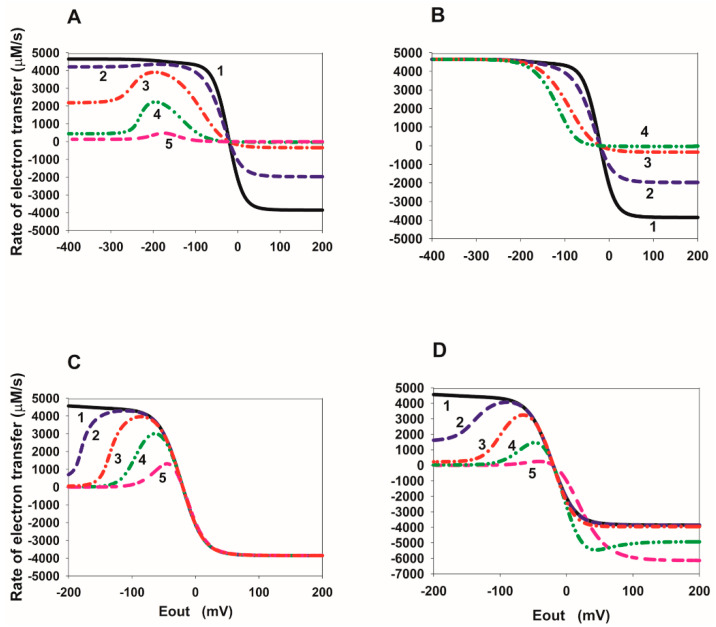
Computer simulated dependence of the steady-state rate of reverse electron transfer in SDH on the electrode potential E_out_ at changes in the rate constants of fumarate and succinate binding to the dicarboxylate binding site upon FAD reduction. (**A**) Simultaneous changes in equilibrium constants K_eq10_ and K_eq12_ of binding of fumarate to the dicarboxylate binding site upon FAD reduction due to a decrease in k_on_ rate constants, k_10_ and k_12_. The values of the reverse k_off_ rate constants k_-10_ and k_-12_ are assumed unchanged and equal to 50 s^−1^. Black solid curve (1) corresponds to k_10_ = k_12_ = 1 µM^−1^ s^−1^, K_eq10_ = K_eq12_ = 2∙10^−2^ µM^−1^; blue dashed curve (2)—k_10_ = k_12_ = 10^−1^ µM^−1^ s^−1^, K_eq10_ = K_eq12_ = 2∙10^−3^ µM^−1^; red dash-dot curve (3)—k_10_ = k_12_ = 10^−2^ µM^−1^ s^−1^, K_eq10_ = K_eq12_ = 2∙10^−4^ µM^−1^; green dash-dot-dot curve (4)—k_10_ = k_12_ = 10^−3^ µM^−1^ s^−1^, K_eq10_ = K_eq12_ = 2∙10^−5^ µM^−1^; pink short-long curve (5)—k_10_ = k_12_ = 10^−4^ µM^−1^ s^−1^, K_eq10_ = K_eq12_ = 2∙10^−6^ µM^−1^. (**B**) Simultaneous changes in equilibrium constants K_eq10_ and K_eq12_ due to an increase in k_off_ rate constants, k_-10_ and k_-12_. The values of k_on_ rate constants, k_10_ and k_12_, are assumed unchanged and equal to 1 µM^−1^ s^−1^. Black solid curve (1) corresponds to k_-10_ = k_-12_ = 50 s^−1^, K_eq10_ = K_eq12_ = 2∙10^−2^ µM^−1^; blue dashed curve (2)—k_-10_ = k_-12_ = 500 s^−1^, K_eq10_ = K_eq12_ = 2∙10^−3^ µM^−1^; red dash-dot curve (3)—k_-10_ = k_-12_ = 5∙10^3^ s^−1^, K_eq10_ = K_eq12_ = 2∙10^−4^ µM^−1^; green dash-dot-dot curve (4)—k_-10_ = k_-12_ = 5∙10^4^ s^−1^, K_eq10_ = K_eq12_ = 2∙10^−5^ µM^−1^. (**C**) Simultaneous changes in equilibrium constants K_eq13_, K_eq14_, and K_eq16_ that describes a decrease in the rate of dissociation of succinate from reduced FAD forms due to a decrease in the rate constants, k_14_ and k_16_. The values of the rate constants, k_−14_ and k_−16_, are assumed unchanged and equal to 0.04 µM^−1^ s^−1^. Black solid curve (1) corresponds to k_14_ = k_16_ = 10 s^−1^, K_eq13_ = 2.4∙10^−4^ µM^−1^, K_eq14_ = K_eq16_ = 250 µM; blue dashed curve (2)—k_14_ = k_16_ = 1 s^−1^, K_eq13_ = 2.4∙10^−3^ µM^−1^, K_eq14_ = K_eq16_ = 25 µM; red dash-dot curve (3)—k_14_ = k_16_ = 10^−1^ s^−1^, K_eq13_ = 2.4∙10^−2^ µM^−1^, K_eq14_ = K_eq16_ = 2.5 µM; green dash-dot-dot curve (4)—k_14_ = k_16_ = 10^−2^ s^−1^, K_eq13_ = 0.24 µM^−1^, K_eq14_ = K_eq16_ = 0.25 µM; pink short-long curve (5)—k_14_ = k_16_ = 10^−3^ s^−1^, K_eq13_ = 2.4 µM^−1^, K_eq14_ = K_eq16_ = 0.025 µM. (**D**) Simultaneous changes in equilibrium constants K_eq13_, K_eq14_, and K_eq16_ due to an increase in the rate constants k_−14_ and k_−16_. The values of the rate constants, k_14_ and k_16_, are assumed unchanged and equal to 10 s^−1^. Black solid curve (1) corresponds to k_−14_ = k_−16_ = 0.04 µM^−1^ s^−1^, K_eq13_ = 2.4∙10^−4^ µM^−1^, K_eq14_ = K_eq16_ = 250 µM; blue dashed curve (2)—k_−14_ = k_−16_ = 40 µM^−1^ s^−1^, K_eq13_ = 0.24 µM^−1^, K_eq14_ = K_eq16_ = 0.25 µM; red dash-dot curve (3)—k_−14_ = k_−16_ = 400 µM^−1^ s^−1^, K_eq13_ = 2.4 µM^−1^, K_eq14_ = K_eq16_ = 25∙10^−3^ µM; green dash-dot-dot curve (4)—k_−14_ = k_−16_ = 4∙10^3^ µM^−1^ s^−1^, K_eq13_ = 24 µM^−1^, K_eq14_ = K_eq16_ = 25∙10^−4^ µM; pink short-long curve (5)—k_−14_ = k_−16_ = 4∙10^4^ µM^−1^ s^−1^, K_eq13_ = 240 µM^−1^, K_eq14_ = K_eq16_ = 25∙10^−5^ µM. The total rate of reverse electron transfer (V_rev_tot_) in SDH is computed as the sum of rates of oxidation of the cluster [2Fe-2S] in reactions 5, 6, 9, 11, 13, 15 and equal to V_rev_tot_ = V_5_ + V_6_ + V_9_ + V_11_ + V_13_ + V_15_. The concentration of fumarate and succinate is equal to 1000 and 50 µM, respectively.

**Figure 3 ijms-23-15596-f003:**
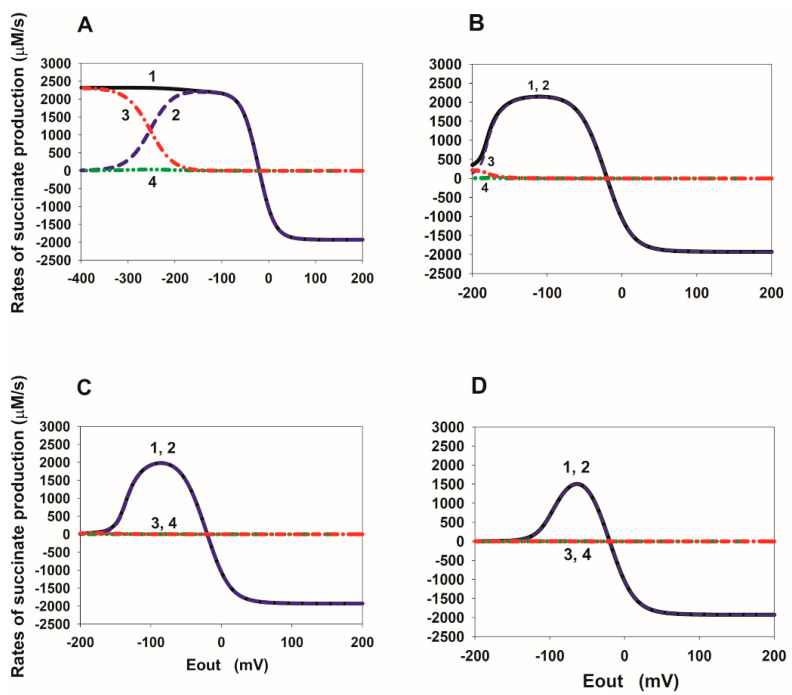
Computer simulated dependence of the steady-state rate of succinate production in SDH on the electrode potential E_out_ at different values of the rate constants of succinate release from the dicarboxylate binding site. The rates of succinate production in reactions 8, 14, and 16 were calculated under simultaneous changes in the values of equilibrium constant K_eq13_ due to proposed succinate-dependent changes in midpoint potential E_m_(FAD.suc/FADH^•^.suc) compared to E_m_(FAD/FADH^•^), as well as equilibrium constants K_eq14_ and K_eq16_ of the release of succinate from the active site upon FAD reduction due to a decrease in the rate constants, k_14_ and k_16_. (**A**–**D**) Black solid curve (1) corresponds to the total rate of succinate production V_suc_tot_ = V_8_ + V_14_ + V_16_; blue dashed curve (2)—V_8_; red dash-dot curve (3)—V_16_; green dash-dot-dot curve (4)—V_14_. All the parameter values for results presented in (**A**–**D**) are the same as for curves (1–4) in Figure 3C. That is, values of k_14_, k_16_, K_eq13_, K_eq14_, and K_eq16_ for the results in (**A**–**D**) are the same as for curve (1), (2), (3), (4) in Figure 2C, respectively, and presented in the Figure 2C caption.

**Figure 4 ijms-23-15596-f004:**
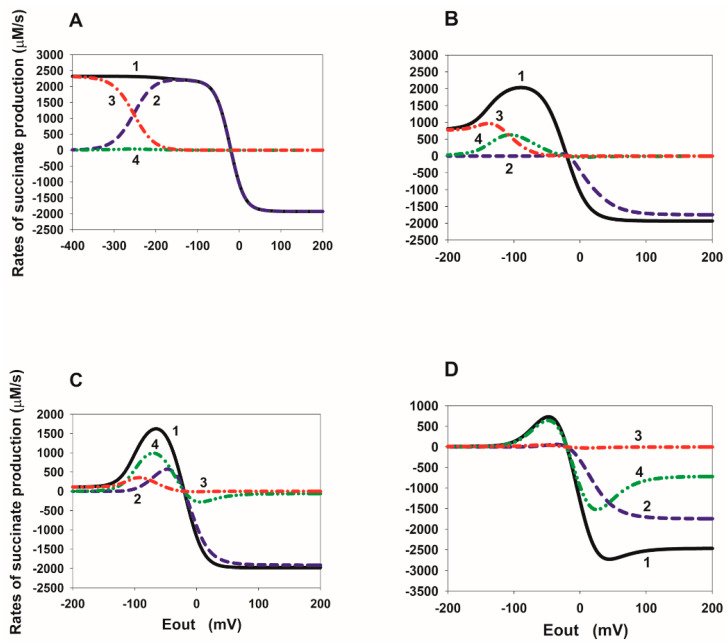
Computer simulated dependence of the steady-state rate of succinate production in SDH on the electrode potential E_out_ at different values of the rate constants of succinate binding to the dicarboxylate binding site. The rates of succinate production in reactions 8, 14, and 16 were calculated under a simultaneous increase in the values of equilibrium constant K_eq13_, as well as a decrease in equilibrium constants K_eq14_ and K_eq16_ of the release of succinate from the active site upon FAD reduction due to an increase in the rate constants k_−14_ and k_−16_. (**A**–**D**) Black solid curve (1) corresponds to the total rate of succinate production V_suc_tot_ = V_8_ + V_14_ + V_16_; blue dashed curve (2)—V_8_; red dash-dot curve (3)—V_16_; green dash-dot-dot curve (4)—V_14_. All the parameter values for results presented in (**A**–**D**) are the same as for curves (1–4) in Figure 2D. That is, values of k_14_, k_16_, K_eq13_, K_eq14_, and K_eq16_ for the results in (**A**–**D**) are the same as for curve (1), (2), (3), (4) in Figure 2D, respectively, and presented in the Figure 2D caption in the main text.

**Figure 5 ijms-23-15596-f005:**
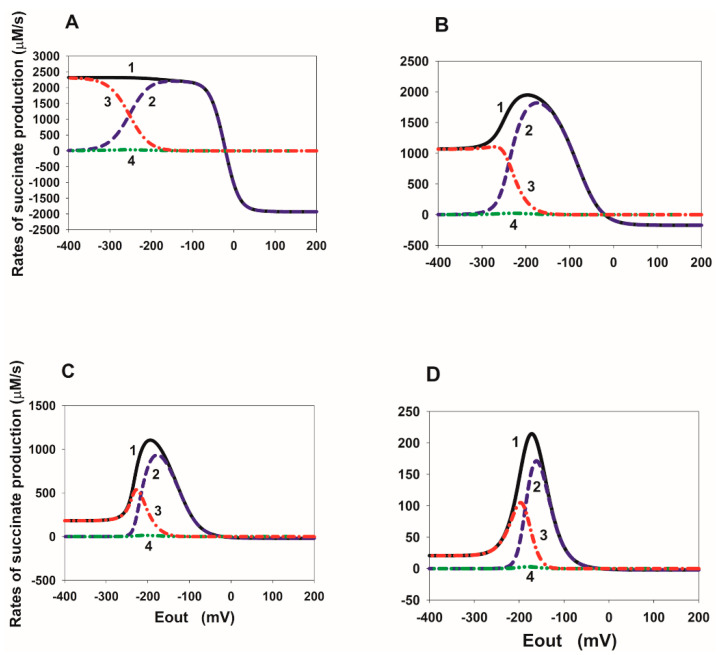
Computer simulated dependence of the stationary rate of succinate production in SDH on the electrode potential E_out_ at different values of the rate constants of fumarate binding to the dicarboxylate binding site. The rates of succinate production in reactions 8, 14, and 16 were calculated under simultaneous decrease in the values of equilibrium constants K_eq10_ and K_eq12_ of binding of fumarate to the active site upon FAD reduction due to a decrease in k_on_ rate constants, k_10_ and k_12_. (**A**–**D**) Black solid curve (1) corresponds to the total rate of succinate production V_suc_ = V_8_ + V_14_ + V_16_; blue dashed curve (2)—V_8_; red dash-dot curve (3)—V_16_; green dash-dot-dot curve (4)—V_14_. All the parameter values for results presented in (**A**–**D**) are the same as for curves (1, 3–5) in Figure 2A. That is, values of k_10_, k_12_, K_eq10_, and K_eq12_ for the results in (**A**–**D**) are the same as for curves (1), (3), (4), (5) in Figure 2A, respectively, and presented in the Figure 2A caption.

**Figure 6 ijms-23-15596-f006:**
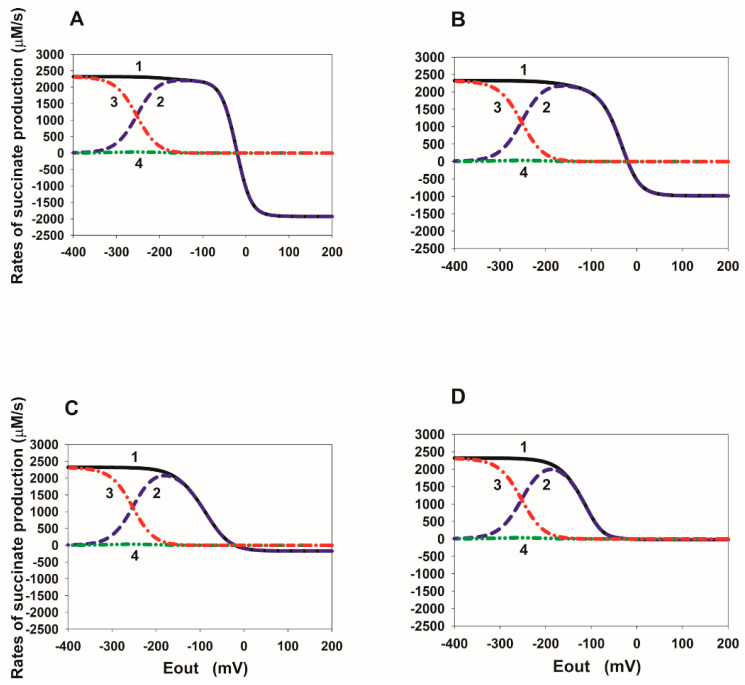
Computer simulated dependence of the stationary rate of succinate production in SDH on the electrode potential E_out_ at different values of the rate constants of fumarate release from the dicarboxylate binding site. The rates of succinate production in reactions 8, 14, and 16 were calculated under simultaneous decrease in the values of equilibrium constants K_eq10_ and K_eq12_ due to an increase in k_off_ rate constants, k_-10_ and k_-12_. The values of k_on_ rate constants, k_10_ and k_12_, are assumed unchanged and equal to 1 µM^−1^ s^−1^. (**A**–**D**) Black solid curve (1) corresponds to the total rate of succinate production V_suc_ = V_8_ + V_14_ + V_16_; blue dashed curve (2)—V_8_; red dash-dot curve (3)—V_16_; green dash-dot-dot curve (4)—V_14_. All the parameter values for results presented in (**A**–**D**) are the same as for curves (1, 3–5) in Figure 2B in the main text. That is, values of k_10_, k_12_, K_eq10_, and K_eq12_ for the results in (**A**–**D**) are the same as for curves (1), (3), (4), (5) in Figure 2B, respectively, and presented in the Figure 2B caption in the main text.

**Figure 7 ijms-23-15596-f007:**
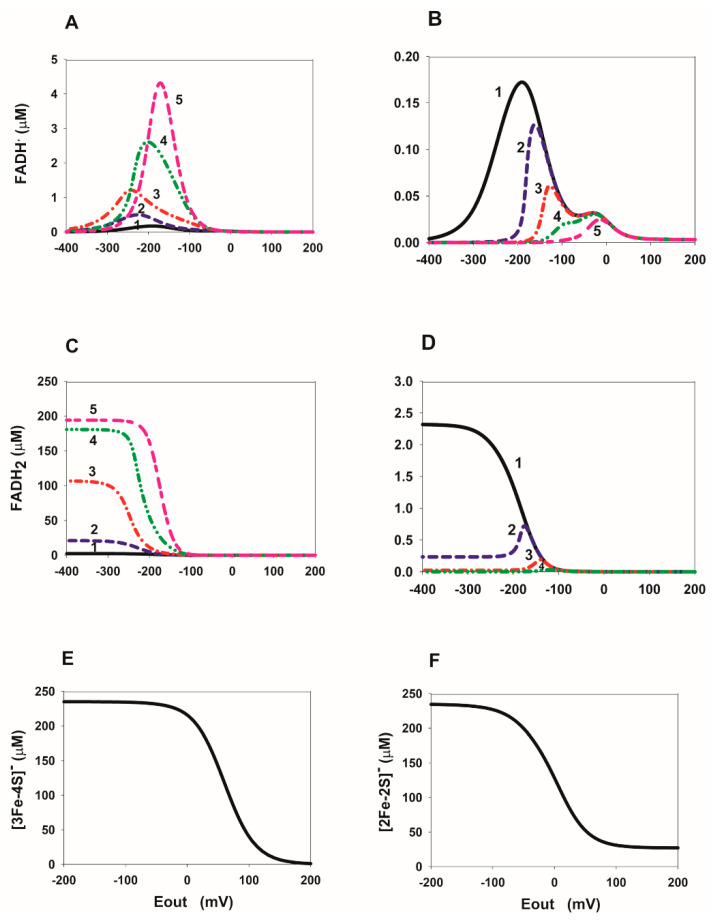
Computer simulated dependence of the concentration of ROS generated redox centers FADH^•^, FADH_2_, and clusters [3Fe-4S] and [2Fe-2S] in the reduced state in SDH on the electrode potential E_out_ at different values of the rate constants of fumarate and succinate binding to the dicarboxylate binding site upon FAD reduction. (**A**,**C**) A dependence of FADH^•^ (**A**) and FADH_2_ (**C**) concentration on E_out_ at a simultaneous decrease in k_on_ rate constants, k_10_ and k_12_, binding of fumarate to the dicarboxylate binding site upon FAD reduction. In this case, the values of the reverse k_off_ rate constants k_-10_ and k_-12_ are assumed unchanged and equal to 50 s^−1^. Black solid curve (1) corresponds to k_10_ = k_12_ = 1 µM^−1^ s^−1^, K_eq10_ = K_eq12_ = 2∙10^−2^ µM^−1^; blue dashed curve (2)—k_10_ = k_12_ = 10^−1^ µM^−1^ s^−1^, K_eq10_ = K_eq12_ = 2∙10^−3^ µM^−1^; red dash-dot curve (3)—k_10_ = k_12_ = 10^−2^ µM^−1^ s^−1^, K_eq10_ = K_eq12_ = 2∙10^−4^ µM^−1^; green dash-dot-dot curve (4)—k_10_ = k_12_ = 10^−3^ µM^−1^ s^−1^, K_eq10_ = K_eq12_ = 2∙10^−5^ µM^−1^; pink short-long curve (5)—k_10_ = k_12_ = 10^−4^ µM^−1^ s^−1^, K_eq10_ = K_eq12_ = 2∙10^−6^ µM^−1^. (**B**,**D**) A dependence of FADH^•^ (**B**) and FADH_2_ (**D**) concentration on E_out_ at a simultaneous decrease in the rate constants, k_14_ and k_16_, of succinate dissociation from reduced FAD forms. The values of the rate constants, k_−14_ and k_−16_ are assumed unchanged and equal to 0.04 µM^−1^ s^−1^. Black solid curve (1) corresponds to k_14_ = k_16_ = 10 s^−1^, K_eq13_ = 2.4∙10^−4^ µM^−1^, K_eq14_ = K_eq16_ = 250 µM; blue dashed curve (2)—k_14_ = k_16_ = 1 s^−1^, K_eq13_ = 2.4∙10^−3^ µM^−1^, K_eq14_ = K_eq16_ = 25 µM; red dash-dot curve (3)—k_14_ = k_16_ = 10^−1^ s^−1^, K_eq13_ = 2.4∙10^−2^ µM^−1^, K_eq14_ = K_eq16_ = 2.5 µM; green dash-dot-dot curve (4)—k_14_ = k_16_ = 10^−2^ s^−1^, K_eq13_ = 0.24 µM^−1^, K_eq14_ = K_eq16_ = 0.25 µM; pink short-long curve (5)—k_14_ = k_16_ = 10^−3^ s^−1^, K_eq13_ = 2.4 µM^−1^, K_eq14_ = K_eq16_ = 0.025 µM. (**E**,**F**) A dependence of the concentration of clusters [3Fe-4S] (**E**) and [2Fe-2S] (**F**) in the oxidized state in SDH on the electrode potential E_out_.

**Figure 8 ijms-23-15596-f008:**
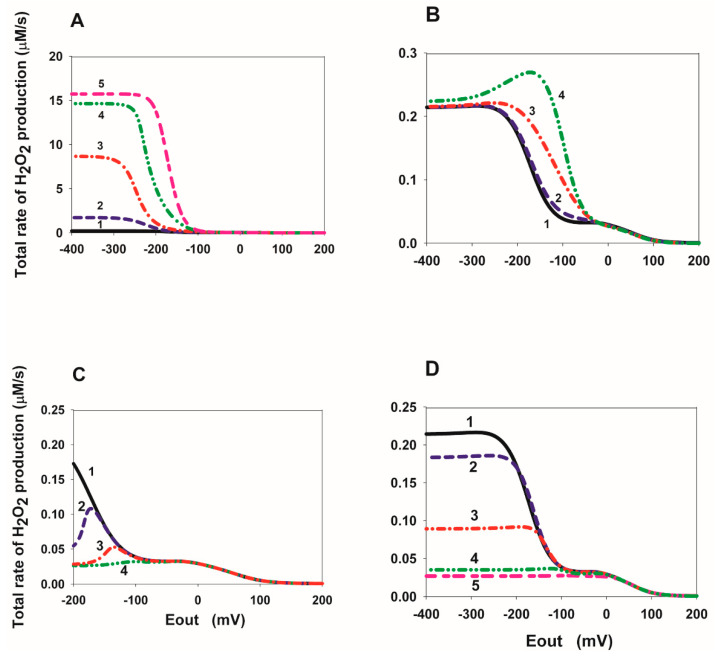
Computer simulated dependence of the total stationary rate of H_2_O_2_ production in SDH on the electrode potential E_out_ at different values of the rate constants of fumarate and succinate binding to the dicarboxylate binding site upon FAD reduction. All the parameter values for results presented in (**A**–**D**) are the same as in Figure 2. That is, values of all parameters for the curves (1–5) in (**A**–**D**) are the same as for curves with the same numbers presented in (**A**–**D**) in Figure 2. (**A**) Simultaneous decrease in equilibrium constants K_eq10_ and K_eq12_ of binding of fumarate to the dicarboxylate binding site upon FAD reduction due to a decrease in k_on_ rate constants, k_10_ and k_12_. In this case, the values of the reverse k_off_ rate constants k_-10_ and k_-12_ are assumed unchanged and equal to 50 s^−1^. Parameter values for curves (1–5) are the same as for curves (1–5) in Figure 2A. (**B**) Simultaneous decrease in equilibrium constants K_eq10_ and K_eq12_ upon FAD reduction due to an increase in k_off_ rate constants, k_-10_ and k_-12_. In this case, the values of k_on_ rate constants, k_10_ and k_12_, are assumed unchanged and equal to 1 µM^−1^ s^−1^. Parameter values for curves (1–4) are the same as for curves (1–4) in Figure 2B. (**C**) Simultaneous changes in equilibrium constants K_eq13_, K_eq14_, and K_eq16_ describe a decrease in the rate of dissociation of succinate from reduced FAD forms due to a decrease in the rate constants, k_14_ and k_16_. The values of the rate constants, k_−14_ and k_−16_ are assumed unchanged and equal to 0.04 µM^−1^ s^−1^. Parameter values for curves (1–4) are the same as for curves (1–4) in Figure 2C. (**D**) Simultaneous changes in equilibrium constants K_eq13_, K_eq14_, and K_eq16_ due to an increase in the rate constants k_−14_ and k_−16_. The values of the rate constants, k_14_ and k_16_, are assumed unchanged and equal to 10 s^−1^. Parameter values for curves (1–5) are the same as for curves (1–5) in Figure 2D. The total rate of H_2_O_2_ production, VH_2_O_2_, was computed as the rate of H_2_O_2_ release from the mitochondrial matrix to the cytosol, V_22_, that equal to the summary rate of H_2_O_2_ production by FADH_2_, V_17_, and dismutation of O_2_∙^−^, V_21_, in the matrix. The concentration of fumarate and succinate is equal to 1000 and 50 µM, respectively.

**Figure 9 ijms-23-15596-f009:**
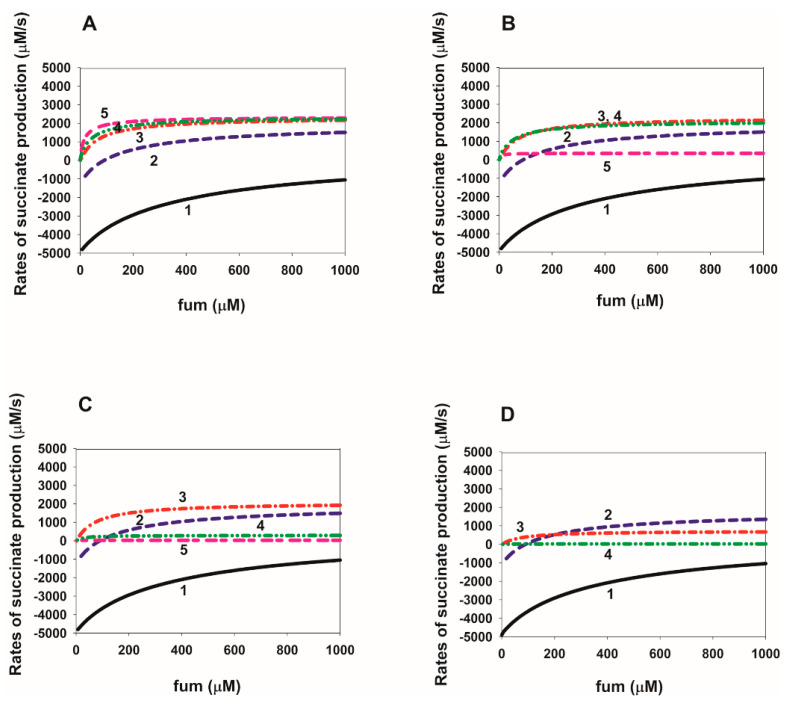
Computer simulated dependence of the stationary rate of succinate production in SDH on the fumarate concentration at different values of the electrode potential E_out_ and rate constants of succinate release from the dicarboxylate binding site upon FAD reduction. (**A**–**D**) Simultaneous changes in equilibrium constants K_eq13_, K_eq14_, and K_eq16_ describe a decrease in the rate of succinate dissociation from reduced FAD forms due to a decrease in the rate constants k_14_ and k_16_. The values of the rate constants, k_−14_ and k_−16_ are assumed unchanged and equal to 0.04 µM^−1^ s^−1^. E_out_ values for black solid curve curves (1) in all figures correspond to 0 mV; blue dashed curves (2) to −50 mV; red dash-dot curves (3) to −100 mV; green dash-dot-dot curves (4) to −150 mV; pink short-long curves (5) to −200 mV. Values of equilibrium constants K_eq13_, K_eq14_ and K_eq16_ and the rate constants, k_14_ and k_16_ are different for Figures (**A**–**D**) and have the following values: (**A**) k_14_ = k_16_ = 10 s^−1^, K_eq13_ = 2.4∙10^−4^ µM^−1^, K_eq14_ = K_eq16_ = 250 µM; (**B**) k_14_ = k_16_ = 1 s^−1^, K_eq13_ = 2.4∙10^−3^ µM^−1^, K_eq14_ = K_eq16_ = 25 µM; (**C**) k_14_ = k_16_ = 10^−1^ s^−1^, K_eq13_ = 2.4∙10^−2^ µM^−1^, K_eq14_ = K_eq16_ = 2.5 µM; (**D**) k_14_ = k_16_ = 10^−2^ s^−1^, K_eq13_ = 0.24 µM^−1^, K_eq14_ = K_eq16_ = 0.25 µM. The succinate concentration is equal to 50 µM.

**Figure 10 ijms-23-15596-f010:**
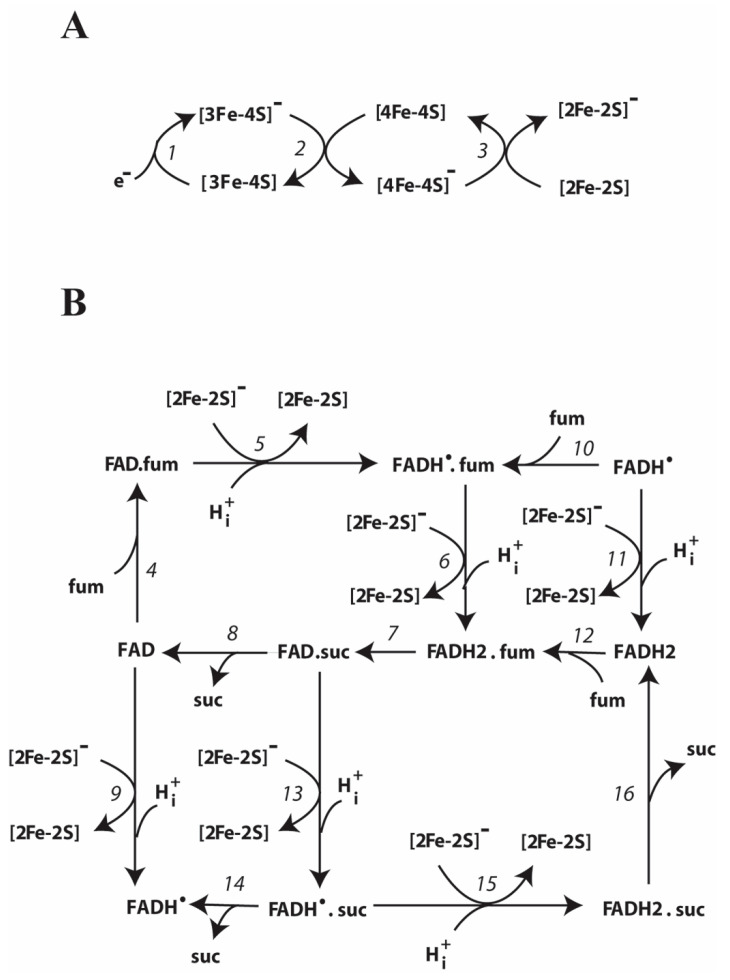
Kinetic schemes of reverse electron transfer and formation of superoxide anion, O_2_∙^−^, and hydrogen peroxide, H_2_O_2_, in the SDHA/SDHB subunits of succinate dehydrogenase (SDH). (**A**) Electron transfer through the three [Fe–S] clusters located in the SDHB subunit. (**B**) Electron transfer and the interconversion of succinate and fumarate in the flavoprotein subunit SDHA. (**C**) Reactions of O_2_∙^−^ and H_2_O_2_ formation in the SDHA/SDHB subunits. The detailed reaction network is presented in Table 1 and Table 2.

**Table 1 ijms-23-15596-t001:** Reactions and rate equations in reverse electron transfer in SDHA/B subunits of SDH.

No	Reaction	Rate Equation
1	e^−^ + [3Fe-4S] = [3Fe-4S]^−^	V_1_ = k_1_ ∙ ([3Fe-4S] ∙ exp(-α ∙ F ∙ E_out_/RT) − [3Fe-4S]^−^ ∙ exp((1-α) ∙ F ∙ E_out_/RT)/K_eq1_)
2	[4Fe-4S] + [3Fe-4S]^−^ = [4Fe-4S]^−^ + [3Fe-4S]	V_2_ = k_2_ ∙ ([4Fe-4S] ∙ [3Fe-4S]^−^ − [4Fe-4S]^−^ ∙ [3Fe-4S]/K_eq2_)
3	[2Fe-2S] + [4Fe-4S]^−^ = [2Fe-2S]^−^ + [4Fe-4S]	V_3_ = k_3_ ∙ ([2Fe-2S] ∙ [4Fe-4S]^−^ − [2Fe-2S]^−^ ∙ [4Fe-4S]/K_eq3_)
4	FAD + fum = FAD.fum	V_4_ = k_4_ ∙ (FAD ∙ fum − FAD.fum/K_eq4_)
5	FAD.fum + [2Fe-2S]^−^ + H ^+^ = FADH^•^.fum + [2Fe-2S]	V_5_ = k_5_ ∙ (FAD.fum ∙ [2Fe-2S]^−^ ∙ H ^+^ − FADH^•^.fum ∙ [2Fe-2S]/K_eq5_)
6	FADH^•^.fum + [2Fe-2S]^−^ + H ^+^ = FADH2.fum + [2Fe-2S]	V_6_ = k_6_ ∙ (FADH^•^.fum ∙ [2Fe-2S]^−^ ∙ H ^+^ − FADH2.fum ∙ [2Fe-2S]/K_eq6_)
7	FADH2.fum = FAD.suc	V_7_ = k_7_ ∙ (FADH2.fum − FAD.suc/K_eq7_)
8	FAD.suc = FAD + suc	V_8_ = k_9_ ∙ (FAD.suc − FAD ∙ suc/K_eq8_)
9	FAD + [2Fe-2S]^−^ + H ^+^ = FADH^•^ + [2Fe-2S]	V_9_ = k_9_ ∙ (FAD ∙ [2Fe-2S]^−^ ∙ H ^+^ − FADH^•^ ∙ [2Fe-2S]/K_eq9_)
10	FADH^•^ + fum = FADH^•^.fum	V_10_ = k_10_ ∙ (FADH^•^ ∙ fum − FADH^•^.fum/K_eq10_)
11	FADH^•^ + [2Fe-2S]^−^ + H ^+^ = FADH2 + [2Fe-2S]	V_11_ = k_11_ ∙ (FADH^•^ ∙ [2Fe-2S]^−^ ∙ H ^+^ − FADH2 ∙ [2Fe-2S]/K_eq11_)
12	FADH2 + fum = FADH2.fum	V_12_ = k_12_ ∙ (FADH2 ∙ fum − FADH2.fum/K_eq12_)
13	FAD.suc + [2Fe-2S]^−^ + H ^+^ = FADH^•^.suc + [2Fe-2S]	V_13_ = k_13_ ∙ (FAD.suc ∙ [2Fe-2S]^−^ ∙ H ^+^ − FADH^•^.suc ∙ [2Fe-2S]/K_eq13_)
14	FADH^•^.suc = suc + FADH^•^	V_14_ = k_14_ ∙ (FADH^•^.suc − FADH^•^ ∙ suc/K_eq14_)
15	FADH^•^.suc + [2Fe-2S]^−^ + H ^+^ = FADH2.suc + [2Fe-2S]	V_15_ = k_15_ ∙ (FADH^•^.suc ∙ [2Fe-2S]^−^ ∙ H ^+^ − FADH2.suc ∙ [2Fe-2S]/K_eq15_)
16	FADH2.suc = suc + FADH2	V_16_ = k_16_ ∙ (FADH2.suc − FADH2 ∙ suc/K_eq16_)
**Hydrogen peroxide (H_2_O_2_) production by SDH**		
17	FADH2 + O_2_ = FAD + H_2_O_2_	V_17_ = k_17_ ∙ (FADH2 ∙ O_2_ − FAD ∙ H_2_O_2_/K_eq17_)
**Superoxide anion (O_2_∙^−^) production by the subcomplex SDHA/SDHB of SDH**		
18	FADH2 + O_2_ = FADH^•^ + O_2_∙^−^ + H ^+^	V_18_ = k_18_ ∙ (FADH2 ∙ O_2_ − FADH^•^ ∙ O_2_∙^−^ ∙ H ^+^ /K_eq18_)
19	FADH^•^ + O_2_ = FAD + O_2_∙^−^ + H ^+^	V_19_ = k_19_ ∙ (FADH^•^ ∙ O_2_ − FAD ∙ O_2_∙^−^ ∙ H ^+^ /K_eq19_)
20	[3Fe-4S]^−^ + O_2_ = [3Fe-4S] + O_2_∙^−^	V_20_ = k_20_ ∙ ([3Fe-4S]^−^ ∙ O_2_ − [3Fe-4S] ∙ O_2_∙^−^/K_eq20_)
**Superoxide anion dismutation in the mitochondrial matrix**		
21	2 O_2_∙^−^ + 2H ^+^ → O_2_ + H_2_O_2_	V_21_ = V_max21_ ∙ O_2_∙^−^/(K_m21_ + O_2_∙^−^)
**Release of hydrogen peroxide (H_2_O_2_) from the mitochondrial matrix to cytosol**		
22	H_2_O_2_ →	V_22_ = k_22_ ∙ H_2_O_2_

**Table 2 ijms-23-15596-t002:** Parameter values for the model.

Reaction No	Midpoint Potential E_m_ = E, (mV)	Equilibrium Constant K_eq_	k_forward_	Other Parameters	Reference
1	E([3Fe-4S]) = 60	11.023	1∙10^3^ s^−1^	α = 0.5 α is a certain coefficient that varies from 0 to 1. E_out_—electrode potential. F, R and T have a usual meaning.	[17] ^a^
2	E([4Fe-4S]) = −260 E([3Fe-4S]) = 60	2.78∙10^−6^	1∙10^4^ µM^−1^·s^−1^	pH = 7.4	[17,18] ^a^
3	E([2Fe-2S]) = 0 E([4Fe-4S]) = −260	3.29∙10^4^	1∙10^4^ µM^−1^·s^−1^	pH = 7.4	[18] ^a^
4		4.17∙10^−3^ µM^−1^	1 s^−1^		[15] ^b^
5	E(FAD/FADH^•^) = −127 E([2Fe-2S]) = 0	0.03 ^e^ µM^−1^	1∙10^3^ µM^−2^·s^−1^	pH = 7 pH = 7.4	[18,19] ^a^
6	E(FADH^•^/FADH2) = −31 E([2Fe-2S]) = 0	0.289 ^e^ µM^−1^	1∙10^3^ µM^−2^·s^−1^	pH = 7 pH = 7.4	[18,19] ^a^
7		2778 ^d^	2.78∙10^6^ s^−1^		
8		10 µM	0.5 s^−1^		[19] ^b^
9	E(FAD/FADH^•^) = −127 E([2Fe-2S]) = 0	0.006	1∙10^3^ µM^−2^·s^−1^	pH = 7 pH = 7.4	[18,19] ^a^
10		0.02 µM^−1^	10^3^ s^−1^		[15] ^b^
11	E(FADH^•^/FADH2) = −31 E([2Fe-2S]) = 0	0.289 µM^−1^	1∙10^3^ µM^−2^·s^−1^	pH = 7 pH = 7.4	[18,19] ^a^
12		0.02 µM^−1^	10^3^ s^−1^		[15] ^b^
13	E(FAD/FADH^•^) = −127 E([2Fe-2S]) = 0	2.4∙10^−4 e^ µM^−1^	1∙10^3^ µM^−2^·s^−1^	pH = 7 pH = 7.4	[18,19] ^a^
14		250 µM	10 s^−1^		[15] ^b^
15	E(FADH^•^/FADH2) = −31 E([2Fe-2S]) = 0	0.289 ^e^ µM^−1^	1∙10^3^ µM^−2^·s^−1^	pH = 7 pH = 7.4	[18,19] ^a^
16		250 µM	10 s^−1^		[15] ^b^
**Hydrogen peroxide (H_2_O_2_) production by Complex II**					
17	E(O_2_/H_2_O_2_) = 690 E(FAD/FADH2) = −79	5.2∙10^26^	0.01 µM^−1^·s^−1^	pH = 7	[19] ^a^
**Superoxide anion (O_2_∙^−^) production by the subcomplex SDHA/SDHB of SDH**					
18	E(O_2_/O_2_∙^−^) = −160 E(FADH^•^/FADH2) = −31	6∙10^−3^	0.01 µM^−1^·s^−1^	pH = 7	[20] ^a^ [19] ^a^
19	E(O_2_/O_2_∙^−^) = −160 E(FAD/FADH^•^) = −127	0.267	0.1 µM^−1^·s^−1^	pH = 7 pH = 7.4	[20] ^a^ [19] ^a^
20	E(O_2_/O_2_∙^−^) = −160 E([3Fe-4S]) = 60	1.5∙10^−4^	1·10^−3^ µM^−1^·s^−1^	pH = 7 pH = 7.4	[20] ^a^ [17] ^a^
**Accompanying reactions in the matrix and inner membrane**					
**Superoxide anion dismutation in the mitochondrial matrix**					
21				V_max21_ = 5.6∙10^4^ µM·s^−1 f^ K_m21_ = 50 µM	[21] ^d^
**Efflux of hydrogen peroxide (H_2_O_2_) from the mitochondrial matrix to cytoplasm**					
22			30 s^−1^		[22] ^c^

^a^ The reference for the midpoint redox potential E_M_. ^b^ The reference for the equilibrium constant K_eq_. ^c^ The reference for the rate constant of direct reaction k_forward_. ^d^ The used value of K_eq7_ = 2778 is calculated from the relation K_eq7_ ∙ K_eq8_ ∙ K_eq12_ = exp (2 ∙ F ∙ (E (fum/suc) − E (FAD/FADH2))/R ∙ T) = 555.6, where midpoint redox potentials E (FAD/FADH2) = −79 mV (pH 7.0) and E (fum/suc) = 0 mV (pH 7.0) [22], respectively, and F, R, and T have the usual meaning. In the general case, when the values of the K_eq8_ and K_eq12_ can vary, K_eq7_ = 555.6/K_eq8_ ∙ K_eq12_. ^e^ Relations between equilibrium constants according to 4 thermodynamic cycles in the kinetic scheme: (1) K_eq5_ = 1/K_eq4_ ∙ K_eq6_ ∙K_eq7_ ∙ K_eq8_; (2) K_eq6_ = K_eq11_ ∙ K_eq12_/K_eq10_; (3) K_eq13_ = K_eq8_ ∙ K_eq9_/K_eq14_; (4) K_eq15_ = 1/K_eq7_ ∙ K_eq12_ ∙ K_eq13_ ∙ K_eq16_; ^f^ The used value was taken from [21], which was calculated from experimental data on Mn–SOD activity in mitochondria of cardiac cells [23]. *Conserved moieties (in µM).* The pool of electron carriers. In this work, we assumed that the concentration of the SDHA/SDHB subunits of SDH is equal to the concentration of Complex II. According to [24], the content of Complex II in cardiac mitochondria is 0.209 nmol Complex II/mg of mitochondrial protein. Translation of whole membrane concentration expressed in nmol/mg mit.prot. to local protein concentration expressed in μM presented in [21]. We have shown earlier [21] that 1 nmol/mg of protein corresponds to 273 μM when normalized to the mitochondrial volume (V_mit_). If the concentration is normalized to the inner mitochondrial membrane volume (V_imb_) it should be additionally taken into account that the ratio W_imb_ = V_imb_/V_mit_ = 0.24 [21]. Therefore 0.209 nmol complex II/mg of mitochondrial protein corresponds approximately to 235 μM if it is recalculated to the concentration in the inner MM (0.209 ∙ 273/0.24 = 237). So, we suggested in the present study that the total concentration of all redox centers localized in Complex II, that is, [FAD], [2Fe-2S], [4Fe-4S], and [3Fe-4S], equal to 235 μM.

## Data Availability

Not applicable.

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
