# Peer review of "Mathematical Modeling of ROS Production and Diode-like Behavior in the SDHA/SDHB Subcomplex of Succinate Dehydrogenases in Reverse Quinol-Fumarate Reductase Direction"

_ijms, 2022, doi:10.3390/ijms232415596_

Round 1

Reviewer 1 Report

Succinate dehydrogenase plays a key role in several physiological processes. Therefore, the topic still has particular interest, even that this complex has been extensively studied previously. 

In line 61, the authors state "Thus, the main goal of this work was not only to analyze kinetic mechanisms of  diode-like behaviour of SDH in reverse quinol-fumarate reductase direction but also to  study conditions when diode-like behaviour is observed." I don't think the authors determine the kinetic mechanism".

The conclusions the authors claim should be proven by Quamtum chemistry, including the dynamics of the system, determining intermediate species, and adjustments in the systems. Then it is possible to determine constant rates, not only based on a fitting mathematical model with few experimental data. 

Author Response

Reviewer #1:

Comments and Suggestions for Authors

Succinate dehydrogenase plays a key role in several physiological processes. Therefore, the topic still has particular interest, even that this complex has been extensively studied previously. 

Comment No.1.

In line 61, the authors state "Thus, the main goal of this work was not only to analyze kinetic mechanisms of  diode-like behaviour of SDH in reverse quinol-fumarate reductase direction but also to  study conditions when diode-like behaviour is observed." I don't think the authors determine the kinetic mechanism".

Response to Reviewer 1 comment No.1

Thank you for this comment. When we wrote that the main goal of this work is to analyze kinetic mechanisms of diode-like behavior of SDH in reverse quinol-fumarate reductase direction, we meant computational modeling analysis of changes in kinetics of electron transfer responsible for diode-like behavior using various kinetic schemes of reverse electron transfer and identifying which of the kinetic pathways is responsible for the diode-like effect. We agree that the more exact concept of the kinetic mechanisms of reactions implies analysis of different parameters like activation energy, pre-exponential factor, order of reaction, reaction type and so on. Therefore, in order not to confuse these concepts, we decided to change this sentence and write “changes in kinetics of electron transfer responsible for diode-like behavior of SDH” instead kinetic mechanisms of diode-like behavior of SDH.

Comment No.2.

The conclusions the authors claim should be proven by Quamtum chemistry, including the dynamics of the system, determining intermediate species, and adjustments in the systems. Then it is possible to determine constant rates, not only based on a fitting mathematical model with few experimental data. 

Response to Reviewer 1 comment No.2

Thank you for this very serious comment. We have no doubt that quantum chemistry could help to study more deeply the mechanisms of changing rate constants leading to the tunnel effect. Unfortunately, the authors have never used quantum chemistry methods to study the behavior of complex biochemical systems. Only mathematical methods of enzymatic kinetics and systems biology, which allow us to estimate the rates of change of fluxes and concentrations of various intermediates in complex systems in time and steady state under various parameter changes. At the same time, deeper changes in biochemical systems using the quantum theory of the structure of molecules, chemical bonds and intermolecular interactions, as well as the quantum theory of chemical reactions and reactivity were not considered.

This work shows that simple methods of mathematical modeling already allow us to evaluate and explain at the behavioral level the experimentally observed diode-like behavior of SDH in reverse quinol-fumarate reductase direction. We are sorry but we did what we could.

Reviewer 2 Report

In the present work, the authors propose a computational kinetic study to clarify the mechanism of succinate dehydrogenase, an important enzyme that is involved in the reverse electron-transfer, taking place in ischemia phenomenon. The conclusions of the investigation are relevant and can be of interest for a broad readership. In particular, it is provided an explanation of the tunnel-diode behavior of the enzyme and that such action is related to a ROS production decrease.

The conclusions are meticulously supported by the results and, for this reason, I support the publication in International Journal of Molecular Sciences. 

I have few suggestions for the authors that, in my opinion, can improve the readability of the manuscript:

1)     The introduction is very informative, with many useful information; however, I would include a sort of general scheme about the described chemical processes in this section, to make easier the understanding of the written part;

2)     I think that the section “Dimension of local and whole mitochondrial concentration and rates.” can be moved to the supporting information file;

3)     Can the authors provide a higher resolution version of the images of the Results and discussion section?

4)     I suggest, in general, a revision of English.

Author Response

Reviewer 2:

Comments and Suggestions for Authors

In the present work, the authors propose a computational kinetic study to clarify the mechanism of succinate dehydrogenase, an important enzyme that is involved in the reverse electron-transfer, taking place in ischemia phenomenon. The conclusions of the investigation are relevant and can be of interest for a broad readership. In particular, it is provided an explanation of the tunnel-diode behavior of the enzyme and that such action is related to a ROS production decrease.

Comment No.1.

The introduction is very informative, with many useful information; however, I would include a sort of general scheme about the described chemical processes in this section, to make easier the understanding of the written part.

Response to Reviewer 2 comment No. 1.

Thank you for this suggestion. We include a simplified general scheme of reverse electron transfer in complex II from quinol QH2 at the CoQ site to the three [Fe-S] clusters located in SDHB subunit and further to the flavoprotein subunit SDHA.

Comment No.2.

 I think that the section “Dimension of local and whole mitochondrial concentration and rates.” can be moved to the supporting information file;

Response to Reviewer 2 comment No. 2.

We agree with you and moved this sectionDimension of local and whole mitochondrial concentration and rates” to the supporting information file (Supplementary Materials).

Comment No.3.

Can the authors provide a higher resolution version of the images of the Results and discussion section?

Response to Reviewer 2 comment No. 3.

Thank you. We have replaced all Figures (1-10) in the manuscript with a higher resolution version (from 300 to 600 dpi) throughout the text.

Comment No.4.

I suggest, in general, a revision of English.

Response to Reviewer 2 comment No. 4.

Thank you for this suggestion. But, unfortunately, I can't change the English syntax of the whole text, only the grammar. I understand that English is not my native language, although I worked for 12 years at Thomas Jefferson University in Philadelphia, USA, and got the skills of scientific English, but apparently not perfect. Although another reviewer recognized the English in this manuscript

 as good enough. Once again, I ask you to forgive me, but I did what I could.

Round 2

Reviewer 1 Report

Change the title to include that this is a mathematical work.

Improve pics quality and labels size.

Author Response

Second Comments and Suggestions for Authors

Comment No.1.

Change the title to include that this is a mathematical work.

Response to Reviewer 1 comment No.1

Thank you. We have changed the old title to the following new one: “Mathematical modeling of ROS production and diode-like behavior in the SDHA/SDHB subcomplex of succinate dehydrogenases in reverse quinol-fumarate reductase direction”

Comment No.2.

Improve pics quality and labels size

Response to Reviewer 1 comment No.2

Thank you. We improved quality of all Figures in the manuscript and labels size.
